# Impact of Metacognitive and Psychological Factors in Learning-Induced Plasticity of Resting State Networks

**DOI:** 10.3390/biology11060896

**Published:** 2022-06-10

**Authors:** Valentine Chirokoff, Georges Di Scala, Joel Swendsen, Bixente Dilharreguy, Sylvie Berthoz, Sandra Chanraud

**Affiliations:** 1Section of Life and Earth Sciences, Ecole Pratique des Hautes Etudes, PSL Research University, 75014 Paris, France; joel.swendsen@u-bordeaux.fr (J.S.); sandra.chanraud@u-bordeaux.fr (S.C.); 2Unité Mixte de Recherche 5287, Centre National de la Recherche Scientifique, Institut de Neurosciences Cognitives et Intégratives d’Aquitaine-Bordeaux University, 33076 Bordeaux, France; georges.di-scala@u-bordeaux.fr (G.D.S.); bixente.dilharreguy@u-bordeaux.fr (B.D.); sylvie.berthoz-landron@inserm.fr (S.B.); 3Psychiatry Unit, Institut Mutualiste Montsouris 42, Boulevard Jourdan, 75014 Paris, France

**Keywords:** resting state, plasticity, learning, confidence, psychological traits, cerebellum

## Abstract

**Simple Summary:**

Connections within the brain can reshape themselves to rapidly adapt to new learning. We aimed to demonstrate that these reconfigurations do not only reflect a memory trace but a more global response to other processes involved in learning. Furthermore, we investigated why individuals do not present the same ability both in learning and in connection plasticity. Present results indicate that brain rapid reconfiguration is not only linked to learning abilities but also to the process of confidence in learning. Factors such as age, education, and anxiety also appear to influence the brain’s response to learning and explain part of the variability observed between subjects. This study revealed important links between brain and psychological functioning and how they influence each other which highlights the need for considering psychological factors both in education and in psychiatric disorders.

**Abstract:**

While resting-state networks are able to rapidly adapt to experiences and stimuli, it is currently unknown whether metacognitive processes such as confidence in learning and psychological temperament may influence this process. We explore the neural traces of confidence in learning and their variability by: (1) targeting rs-networks in which functional connectivity (FC) modifications induced by a learning task were associated either with the participant’s performance or confidence in learning; and (2) investigating the links between FC changes and psychological temperament. Thirty healthy individuals underwent neuropsychological and psychometric evaluations as well as rs-fMRI scans before and after a visuomotor associative learning task. Confidence in learning was positively associated with the degree of FC changes in 11 connections including the cerebellar, frontal, parietal, and subcortical areas. Variability in FC changes was linked to the individual’s level of anxiety sensitivity. The present findings indicate that reconfigurations of resting state networks linked to confidence in learning differ from those linked to learning accuracy. In addition, certain temperament characteristics appear to influence these reconfigurations.

## 1. Introduction

### 1.1. Behavioral Relevance of Resting State Networks Connectivity

The “resting” brain is in fact highly active. In the absence of cognitive tasks or external stimulation, Blood Oxygen Level Dependent (BOLD) signals demonstrate widespread spontaneous fluctuations that are likely to reflect the brain’s intrinsic functional architecture. The temporal correlations between BOLD signals from different brain regions are not random and reflect functionally relevant resting state (rs) networks. These networks can be reliably observed across individuals and neuroimaging sessions [1], with small between-person variation in connectivity patterns and strength [2]. Between-group comparisons of rs-functional connectivity organization accurately allow for the identification of pathological biomarkers [3,4], and Rs connectivity predicts task activation patterns [5] and performance in a variety of cognitive tasks [6,7].

Despite this relatively stable organization, rs-networks also exhibit short-term [8] and long-term plasticity [9]. In particular, these networks can be modified not only by injury or lesion, but also in response to practice or experiences [8]. Such task-induced changes in functional connectivity (FC), either by reconfiguration or by changes in the strength of connections, offer new possibilities to investigate higher-order cognitive processes. At the same time that investigations of these experience-induced modifications progressed, methodologies and guidelines were proposed to study these modifications from either psychological or neurobiological perspectives. To date, however, only a few studies have combined both approaches.

Kelly and Garavan [10] presented a framework to interpret practice-related changes in the human brain and indicated the importance of first differentiating redistribution (an increase and/or decrease in activation within the same areas during the task) from a “true” reorganization (a change in activation location). These authors also emphasized the necessity of carefully considering the behavioral and cognitive operations involved during the task when interpreting changes within or between brain areas and they specified the particular importance of three factors: (i) the effect of practice, where the interpretation of changes within the brain must integrate changes in the cognitive processes underlying the performance of the task; (ii) the effect of task domain, and more specifically in terms of high-level cognitive processes relative to low-level perceptual processes or motor skills; and (iii) the effect of neuroimaging time windows, whereby researchers must consider the entire timeframe over which practice-related effects may occur. Importantly, these authors concluded that connectivity analyses are particularly informative and allow researchers to overcome multiple limits encountered while studying activation analyses [8].

### 1.2. RS-Networks and Learning Traces

Several authors have previously studied the modification of rs-functional connectivity induced by motor learning tasks. For instance, Albert et al. [11] reported increased connections within frontoparietal and cerebellar rs-networks among healthy participants who performed a motor adaptation learning task in comparison to a simple motor performance task. Other studies, which used a similar motor or sensorimotor associative learning task coupled with pre- and post-task resting fMRI acquisitions, also identified connectivity strength modifications within multiple networks. These changes were identified within the frontoparietal [12,13], fronto-cerebellar circuits [12,13,14,15], as well as within the cerebellum [13,14]. These changes have been considered to reflect the “off-line” processing of learning with task-specific rs-networks supporting memory consolidation [11,12,13,14,15,16]. For Bassett and colleagues, complex cognitive processes such as behavioral adaptation or learning rely on a continuous evolution of functional connections [17]. In support of this hypothesis, modulation of rs-networks has been found to correlate with learning performance [16,18].

However, it is still unclear why and how practice-induced changes in specific rs-networks vary across subjects. During the past few years, the implication of metacognitive abilities has been increasingly discussed, notably self-confidence which is considered to be a core component of the learning process [19]. Confidence in learning is essential for the guidance of behavior [20] and for probabilistic learning in the human brain [21], and may thus influence memory consolidation. Although it has been suggested that confidence is linked to task-based networks fluctuations [22], to date this question has not been investigated experimentally using rs-networks. Importantly, even among non-clinical populations, large inter-individual variability has been observed in confidence ratings [23] and such heterogeneity has been hypothesized to originate from, or to be reflected in, brain organization [24,25]. Moreover, psychological factors such as anxiety proneness are known to be linked not only to learning but also to inter-individual variability in confidence ratings [26] and may thereby influence confidence-related Rs brain networks [27,28].

In order to address these issues, the present investigation explores the neural traces of confidence in learning and their variability by first targeting rs-networks in which modifications induced by a learning task correlate with the level of confidence of participants. The links between these modifications and psychological dimensions are then be examined. To achieve these objectives, the study design combines neuropsychological and psychometric evaluations, behavioral performance with its associated level of confidence, and resting state fMRI acquisitions before and after a learning task.

## 2. Material and Methods

### 2.1. Sample Size and Power Considerations

Group size was determined based on power curves obtained for cognitive and resting state fMRI experiments. Although these curves suggested that for a liberal threshold of 0.05, about 12 subjects are required, a sample size superior to 30 is based on a more conservative estimation of these curves in order to have sufficient power to address secondary questions.

### 2.2. Participants

Subjects were recruited through community announcements if they had no history of neurological or psychiatric illness, had normal or normal-corrected vision, right-handedness and were aged between 18 and 70 years. The study was conducted in agreement with ethical standards of the Helsinki declaration. Subjects were volunteers, did not receive any monetary or material compensations and provided written informed consent for participation.

### 2.3. Psychological Evaluations

Before the fMRI exam, the participants completed the following self-report questionnaires: the Hospital Anxiety and Depression scale (HAD, [29]); the State-Trait Anxiety Inventory (STAI-Y A and B; [30]); the subscales of personal standards, doubts about actions and concern over mistakes from the Frost Multidimensional Perfectionism Scale (FMPS, [31]), and the sensitivity to reward and punishment questionnaire (SPSRQ; [32]).

### 2.4. MRI Acquisition

#### 2.4.1. Learning Task

Participants completed a modified version of Balsters and Ramnani’s conditional visuomotor learning task [33]. During this task, the subject had to learn sets of first order rules, each one corresponding to the arbitrary association between a total of eight randomly presented geometrical shapes and one of three buttons of an MRI-compatible response box. The subject’s right-hand fingers were positioned on the three-button MRI compatible response boxes. The experiment was fully computerized using the E-prime v3.3 software (Psychology Software Tools, Sharpsburg, MD, USA.). As displayed on Figure 1, each trial began with the presentation of a fixation cross (500 ms), immediately followed by the stimulus presentation (500 ms) itself followed by a « Go! » signal (250 ms). Three dashes were then presented (1000 ms) indicating to the subject that the response was needed. Once the subject had answered, a green dot feedback indicated a correct answer, whereas a red dot feedback indicating a wrong answer (250 ms). The experiment stopped either when the subject reached 80% of correct answers or after 300 trials.

Before performing the task in the MRI, the participants were informed of the structure of the task and had a practice session with a reduced version encompassing 16 trials but using stimuli of different shapes and colors than those of the experimental task. We ensured that all participants were able to explicitly describe the general rule of the task before entering the MRI. While being in the scanner, after completing the learning task and before the second resting state acquisition, participants had to rate their level of confidence in their learning skills using a graduated visual analogical scale (from 0% to 100% of confidence).

#### 2.4.2. MRI Acquisition

Participants went through all MRI acquisitions during a single session. Brain imaging data were collected using an MRI 3 Tesla Prisma Siemens. Anatomical MRI volumes were acquired using a sagittal three-dimensional T1-weighted (Repetition Time = 8.2 ms, Echo Time = 3.5 ms, FOV = 256 mm × 256 mm, voxel size = 1 mm × 1 mm × 1 mm, Slice Thickness = 1 mm, 180 slices). Rs functional images were collected using multiecho-planar sequence (RT = 1 s, ET = 30 ms, FOV = 220 mm × 220 mm, voxel size = 2.5 mm × 2.5 mm × 2.5 mm, 60 axial slices, 360 dynamics). All acquisitions were aligned on the AC–PC plane. All subjects performed two resting-state scans for 11 min each: one before the task acquisition run (pre-task rest), and the other after the task acquisition run (post-task rest). During the resting-state fMRI scan sessions, subjects were instructed to stay still with their eyes closed without falling asleep and to not think about anything in particular. Before data processing, MRI images were visually inspected in order to remove those which presented major artifacts (deformations and movements).

#### 2.4.3. Functional Imaging Pre-Processing

Functional MRI data were preprocessed using FreeSurfer and SPM12 (Conn Toolbox). Rs-FC analyses were carried out using the CONN toolbox version 16 [34] implemented in Statistical Parametric Mapping Software (SPM12, https://www.fil.ion.ucl.ac.uk/spm/software/spm12/, accessed on 1 December 2019) on MATLAB 2019.b, (The MathWorks Inc., Natick, MA, USA, http://www.mathworks.fr/products/matlab/, accessed on 3 December 2019). Anatomical images were segmented into grey matter, white matter, and cerebrospinal fluid, using the Computational Anatomy Toolbox (CAT12, http://www. neuro.uni-jena.de/cat/, accessed on 1 June 2018) implemented in SPM12 in order to create masks for BOLD signal extraction.

The functional images were first corrected for differences in slice acquisition timing within each volume using the middle slice as reference. Functional volumes were then realigned to the middle dynamic scan and co-registered to the T1-weighted anatomical image. The Artifact Detection Tools (http:/www.nitrc.org/projects/artifact_detect, accessed on 15 April 2018) was also used to define the temporal confounding factors due to head motion (six rigid-body head motion parameter values—*x*, *y*, *z* translations and rotations). Excessive head motions were verified, and subjects exhibiting movements greater than 2 mm or 2° for each axis were excluded. These temporal confounding factors were then treated as nuisance variables in the general linear model and regressed out of the resting-state time series of each voxel in the grey matter for each subject. Next, images were wrapped into the Montreal Neurological Institute space with grey matter transformation parameters derived from the T1 segmentation; then, the images were finally smoothed with a 8 × 8 × 8 FWHM filter and linearly detrended. To account for cardiac and respiratory artifacts in the resting-state signal, the anatomical component-based noise correction method [35] was used. This method involves extracting signals from white matter and cerebrospinal fluid regions using principal component analysis and then regressing these signals out of the total fMRI signal. Functional volumes were then band-pass filtered (0.009 Hz ≤ *f* ≤ 0.08 Hz) to ensure that analyses were completed within the frequency band of interest after the regression of the confounding variables. Finally, functional images were despiked, which minimizes the effect of outliers in the times-series signal. In order to maintain the same number of dynamics for each subject, removed slices were replaced by the average of *n* − 1 and *n* + 1 slices.

#### 2.4.4. Whole Brain ROI to ROI Functional Connectivity

For each subject, the automated anatomical labeling atlas [36,37] was used to build the regions of interest (ROI) functional connectivity maps. To limit the partial volume effect, each ROI was restricted to voxels belonging to an estimated grey-matter mask derived from the T1-weighted image segmentation. The ROI-to-ROI analysis was carried-out separately for the two rest periods, and for each subject. Mean fMRI time-series were extracted within each ROI. FC was estimated using Pearson’s correlation coefficients between the BOLD time courses of all ROI pairs in each two rest runs (pre- and post-task rest) for each subject. These correlation coefficients (*r*) were finally Fisher *Z*-transformed to produce a 116 × 116 matrix representing the intensities of brain functional connections among the final 116 ROIs.

### 2.5. Statistical Analyses

#### 2.5.1. Rs Functional Connectivity

From the total sample of 37 subjects, 1 participant was excluded due to missing functional data, and another due to excessive movements during the scan, leaving a sample of 35 participants (16 men) with fMRI data. However, five subjects had missing confidence ratings, resulting in a sample of thirty subjects for the Rs analysis. A “learning accuracy network” was identified by applying user-defined contrasts in the Conn Toolbox with rest as a within-subject variable and performance as between-subjects variable. This contrast is equivalent to conducting regression analyses using the level of learning accuracy as the predictor variable and the level of FC changes as the outcome variable. We replicated the contrast with the confidence level to identify a “confidence network”. The after–before difference in the connectivity strength (ΔFC) and the strength of connections during the first Rs of the identified network were then extracted. All results reported were corrected for multiple comparisons using a seed-level false discovery rate (FDR) threshold with an alpha level of 0.05.

#### 2.5.2. Psychological Variables

Given the number of psychological dimensions, and to minimize the risk of Type 1 errors while testing the associations between the ΔFC and the psychological scores, a dimensionality reduction was first performed using a Principal Component Analysis (PCA) with promax rotation and Kaiser normalization (Kappa = 4; maximal iterations = 25). Two factors emerged and their corresponding factorial scores were extracted. Pearson correlations were used to assess the associations between the ΔFC, age, education and factorial scores; first without and then with controlling for the level of accuracy (i.e., simple and partial correlations).

#### 2.5.3. Post-Hoc Analyses

We selected psychological factors that were significantly correlated with ΔFC and investigated the impact on the baseline level of connectivity. Thus, partial Pearson correlations were used to assess the associations between the strength of connectivity before the task (at baseline) and psychological factors while controlling for age and education level. These analyses were performed using JAMOVI software (the jamovi project (2020). *jamovi* (Version 1.2), Sydney, Australia, https://www.jamovi.org, accessed on 4 June 2020) and SPSS (IBM Corp. Released 2017. IBM SPSS Statistics for Windows, Version 25.0. IBM Corp., Armonk, NY, USA). All statistical tests were two-sided and results were considered significant when *p* value was equal or less than 0.05.

## 3. Results

As displayed in Table 1, 30 participants (14 men, total mean age 34.7 ± 13.8) with relatively high education (3.6% with less than a baccalaureate degree, 10.7% with 2 years after the baccalaureate degree and 82.1% with at least 3 years of education after the baccalaureate degree) were included in analyses of task-induced changes in Rs connectivity. For correlational analyses with the psychological factorial scores, a total of 26 participants (12 men; total mean age 34.80 ± 14) with relatively high education (4% with less than a baccalaureate degree, 12% with 2 years after the baccalaureate degree and 84% with at least 3 years of education after the baccalaureate degree) were included due to missing data for certain self-report measures (see Table 1).

### 3.1. Task-Induced Resting State Network Modifications

Two different patterns of changes in functional connectivity (ΔFC) were associated with learning and with confidence. Positive associations between ΔFC and learning accuracy were identified in six pairs of areas, connections here refer to as “learning accuracy network”. Increased connectivity was found between the left inferior frontal orbitary cortex and cerebellar vermis IV/V, between the right amygdala and right cerebellar lobule IX and, between the left superior parietal cortex and left cerebellar lobule III. Decrease in connectivity was found between the right lingual cortex and right cerebellar lobule IV/V, between the left cerebellar lobule III and right cerebellar lobule VIII and right precentral cortex (see Appendix A).

Positive associations between ΔFC and the level of confidence in learning are indicated in red in Figure 2 and were identified in 11 pairs of areas, connections referred to as “confidence network”. As displayed in Figure 2, four of these connections are located between the posterior cerebellum and frontal (left superior orbitary) areas, and subcortical areas including the amygdala and hippocampus; two connections concerned areas within the posterior cerebellum (areas IX and VI). Three subcortico-cortical connections were identified between the right amygdala and the bilateral angular and right precuneus, and between the right angular and left putamen. Finally, one cortico–cortical connection was identified between the right angular and right frontal inferior opercular. All these connections exhibited a decrease in connectivity after the task in comparison to before, the only exception was the connection between the right and left cerebellar lobules IX which exhibited an increase in connectivity after tasking.

### 3.2. Associations between the ΔFC in the Confidence Network and the Psychological Dimensions Examined through Principal Component Analysis (PCA)

Sampling adequacy was acceptable, as suggested by the Barlett’s Test of Sphericity (*p* < 0.001) and the overall Kaiser–Meyer–Olkin test (KMO = 0.77), confirming the suitability for dimensionality reduction. Based on parallel analysis and eigenvalues above 1, two factors accounting for 64% of the total variance were identified. Loading values for each factor are presented in the Appendix A. Factor 1 was mainly composed by all the anxiety scores (STAI trait and state, HAD Anxiety) and the sensitivity to punishment (SPSRQ) score. It accounted for 46% of the variance and was termed “Anxiety sensitivity”.

Factor 2 was mainly composed by the perfectionism (FMPS) subscale scores, the sensitivity to reward (SPSRQ) score as well as the HAD depression score. It accounted for 18% of the variance as was termed “Achievement sensitivity”. For both factors, the individual’s factorial score was extracted for the correlational analyses.

On overall, age was negatively linked to ΔFC of five connections within the confidence networks: right cerebellar Crus II–Left frontal superior orbitary gyrus (r = −0.473); right angular gyrus–right frontal inferior opercular cortex (r = −0.363); right amygdala–right cerebellar lobule IX (r = −0.426); right amygdala–right precuneus (r = −0.369); Left cerebellar lobule IX–right cerebellar lobule IX (r = −0.384). Education level was positively linked to ΔFC in two of these connections: right amygdala–left cerebellar lobule IX (r = 0.473) and right hippocampus–right cerebellar lobule IX (r = 0.428) (see Appendix A).

As illustrated in Figure 3a, the ΔFC of seven of the connections within the confidence network correlated negatively with the anxiety sensitivity factor while controlling for age and education: right angular–right inferior opercular (r = −0.463); right amygdala–left cerebellum IX (r = −0.551); right amygdala–right angular (r = −0.466); right amygdala–left angular (r = 0.517); left putamen–right angular (r = −0.486); left cerebellum IX–right IX (r = −0.485) and right cerebellum IX–Right VI (r = −0.499) (See Appendix A). All these associations remained significant when controlling for level of accuracy except the association with right amygdala–right angular (see Appendix A). With regard to the achievement sensitivity factor, only negligeable associations (<0.20) were found.

### 3.3. Post Hoc Analyses

As illustrated by Figure 3b, the baseline connectivity strength (during first rs) for all connections within the confidence network were negatively correlated with level of confidence. Moreover, FC in two of these connections, the right amygdala–right angular and right cerebellar lobule VI–right cerebellar lobule IX, were positively correlated with anxiety sensitivity factorial score while controlling for age and education (see Appendix A).

## 4. Discussion

In the present study, we assessed changes in resting state architecture induced by associative learning and examined how they relate either to learning performance or to the level of confidence in learning. We observed a redistribution of the functional networks [10], and more precisely a change in connectivity strength between several pairs of areas. Importantly the highlighted learning and confidence networks involved separate functional connections. In accordance with previous studies, we observed task-induced modifications within frontoparietal [12,13], fronto-cerebellar circuits [12,13,14,15], and within the cerebellum [13,14], thus confirming that cognitive processes could leave neural traces in rs-networks.

In addition to these findings, we were able to separate the modifications related specifically to learning performance from those related to confidence in learning. The “learning accuracy network” included cerebellar (lobule III, IV, V, VIII, IX and vermis IV/V), frontal (left inferior orbitary, right precentral), occipital (right lingual), parietal (left superior), and subcortical (right amygdala) areas. These areas had already been involved in different kinds of learning tasks, from basic conditioning to associative learning [13,33,38,39]. The “confidence network” included cerebellar (Crus II, lobule VI and IX), frontal (right inferior opercular, left superior orbital, parietal (bilateral angular, right precuneus) and subcortical (right hippocampus, right amygdala and left putamen) areas. These networks only overlapped on one pair of connections, between the right amygdala and the right cerebellar lobule IX, suggesting a segregation between both neural traces. Concerning the confidence network, most of the identified areas (posterior cerebellum, angular gyrus, precuneus, putamen, amygdala, hippocampus) have already been shown to be part of the Default Mode Network (DMN), usually defined as an introspective and self-referential network [40] and that is closely related to metacognition [41].

Using a memory task and asking participants to rate their confidence in their memories, Ren and colleagues [41] showed that the modulation of effective connectivity between the hippocampus and the precuneus (areas that were also identified in the present study) significantly predicted the individual’s subsequent confidence but not their performance accuracy [41]. Parietal areas are involved not only in the storage of memory representations needed when rating confidence but also in the neural processes themselves underlying metacognitive memory judgement [41,42]. The precuneus and the angular gyrus act as network hubs where multi-sensory information is integrated [41,43], giving rise to several phenomena including manipulation of mental representations [43] and the subjective experience of memory [44].

In addition to memory judgement, some studies suggest that learning and confidence could arise from a common inference process [19] where differences between expected and actual outcomes drive behavioral adaptation. The areas identified in our study have been previously linked to the generation of prediction, and confidence about this prediction, through their connections with other cortical and subcortical areas [19,45,46,47]. Notably, the involvement of the amygdala in emotions is widely recognized but could also be involved in action-selection processes underlying decisions [48] especially due to its involvement in reward-related signals [46]. Similarly, both the putamen and the inferior frontal gyrus identified here have been linked to anticipation and prediction errors during the formulation of confidence judgements, even when external reward-based signals are missing [21,45,47].

Our findings are also consistent with the involvement of the cerebellum in the establishment of internal models. Internal models from the cerebellum participate in learning and automatization in order to allow actions to be performed smoothly and rapidly [49]. Beside motor acquisition and regulation, an increasing number of studies point to the implication of the cerebellum in cognition and emotions [50]. In the present study, more than half of the identified connections included cerebellar areas belonging to the executive or the DMN [51]. While engagement of cerebellum is not surprising given that we used a task known to involve this region [33], it is noteworthy that the majority of the connections which support either performance or confidence included distinct cerebellar regions. Models of metacognition assume that individuals build internal models of both the world and their own cognition to regulate and monitor behavior [52,53]. The process of prediction and comparison of outcomes may even be the crucial component of metacognition [54]. Therefore, the increase in connectivity within cerebello–cerebellar connections in confidence networks could indicate the involvement of internal models in the regulation of metacognition, just as it has been postulated for learning [49], regulation of cognition [55] and regulation of emotion [50].

It should be noted that participants rated their level of confidence just before the second rs-scan, we cannot exclude the possibility that present results might have been different if confidence levels were assessed after the rs-scan. In a similar manner, psychometric questionnaires were completed before the task, which could have an impact on subsequent performance. Despite these limitations, investigation of this short-term plasticity allowed us to identify the neural traces of metacognitive process that seem to rely, at least partially, on connections between cortico or subcortical areas and the cerebellum. Regarding factors of inter-individual variability, our findings revealed that besides age and education, the degree of connectivity change within the confidence network was also linked to psychological traits. Namely, the higher the level of anxiety sensitivity, the lower was the tendency to exhibit connectivity changes after, compared to before, the task. This result was significant for seven connections between the right angular and the right frontal inferior opercular cortex, the right amygdala and the cerebellum (left IX) and the angular (bilateral), between the left putamen and the right angular and within the cerebellum (left IX–right IX and right VI). Importantly, this relation is independent of objective performance (i.e., learning accuracy). Anxiety has been related to measures of cognitive inflexibility [56] and to poor cognitive flexibility during decision-making [57]. This “inflexibility” could also be reflected in rs-FC and could lead to, or be caused by, from brain rigidity as demonstrated by reduced changes during the task. However, further investigation is necessary for validating or refuting this hypothesis.

Nonetheless, based on this finding, we decided to test whether anxiety had an impact on the specified network before performing the task (at the baseline). Our analyses indicate a positive link between anxiety scores and the strength of two connections before the task, between the right amygdala and the right angular and within the cerebellum and between the cerebellum (left IX–right VI). As we had not hypothesized a priori that more anxious participants would exhibit higher connectivity in the targeted network before the task, replication is needed before drawing firm conclusions about a link between anxiety, hyperconnectivity before tasks and task-induced reinforcement in this network. That being said, the relationship between high trait anxiety and hypo- or hyper-connectivity is currently debated in the literature [58]. For instance, hyper-connectivity between the amygdala and the angular has been demonstrated in social anxiety disorder patients compared to controls [59]. In the same way, hyper connectivity between cortical areas and the cerebellum characterizes anxiety disorder patients compared to healthy controls [56,60] and it is also linked to higher trait anxiety in the healthy population [58]. However, as this connectivity is plastic over time, the time windows during which Rs are captured seems to be of primary importance especially as less anxious participants exhibited more changes in the network linked to confidence (and thus more connectivity). Further investigations are needed to test whether short-term plasticity of rs-networks could partially explain the contradictory results in the literature concerning the involvement of hypo or hyper connectivity within these networks [58] and to investigate a possible link between FC at baseline and plasticity of rs-networks.

In conclusion, the present study underscores several important issues to consider on this subject. First, as noted by Kelly and Garavan [10], the interpretation of short-term plasticity such as we investigated must rely on interpretations of underlying changes in cognitive processes. To our knowledge, our study is the first to demonstrate that confidence in learning, and not only learning performance, can leave a trace in resting state networks. Such “neural traces” do not only reflect memory consolidation processes as previously hypothesized [12,13,14,15] but also higher-order latent metacognitive processes involved in learning. Second, we were able to describe involvement of new areas in metacognitive processes. Besides connections belonging to the DMN, we identified several cortico-subcortical loops, and more specifically those including the cerebellum which are reinforced during learning and linked to confidence ratings. These findings are in agreement with the conceptualization that cerebellar regulation plays a more global role that extends far beyond motor control, and impact cognitive and emotional processes [59,60]. Finally, Rs plasticity varies between individuals and seems to be linked to anxiety or anxiety-related factors such as sensibility to punishment. While we are not able to conclude whether such psychological factors arise from brain network rigidity or whether they cause it, the present study emphasizes the importance of taking into account such sources of variability. Future investigations of Rs short-term plasticity should offer new understandings of behavioral and cognitive adaptation and its relation to the human brain.

## Figures and Tables

**Figure 1 biology-11-00896-f001:**
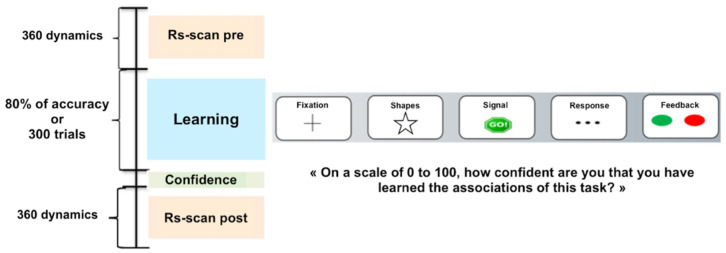
Experimental design.

**Figure 2 biology-11-00896-f002:**
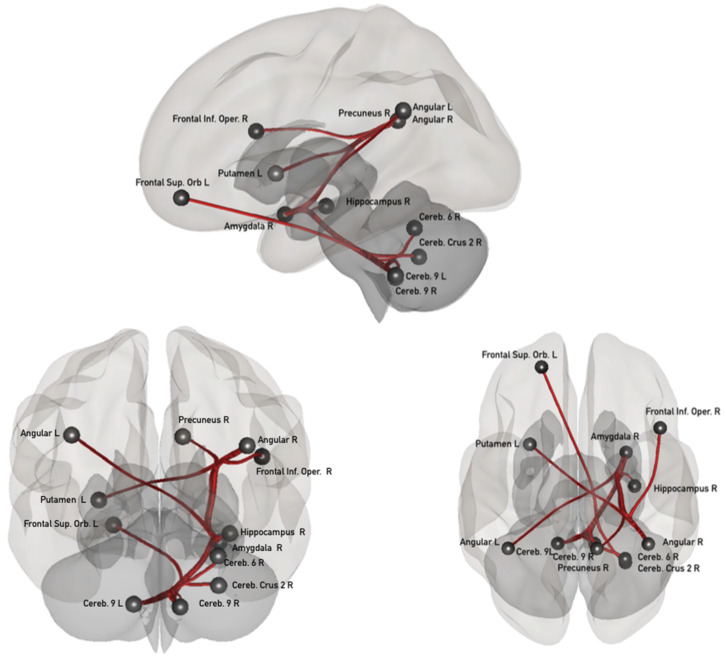
Sagittal, coronal and axial views of the confidence network. L: left; R: right; Inf: inferior; Sup: superior; Cereb.: cerebellum; Orb: orbital; Oper: opercular.

**Figure 3 biology-11-00896-f003:**
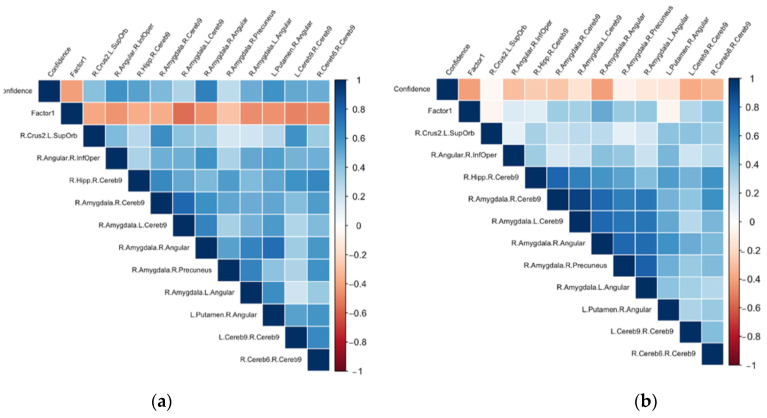
Pearson’s correlations (r) between the confidence level, anxiety sensitivity factorial score and ΔFC (**a**) or strength of connectivity during the first-rs (**b**) while controlling for age and education level. L: left; R: right; Inf: inferior; Sup: superior; Cereb.: cerebellum; Orb: orbital; Oper: opercular; Hipp: hippocampus; Factor 1: “Anxiety sensitivity”.

**Table 1 biology-11-00896-t001:** Descriptive statistics of the behavioral and psychological variables of interest.

	**N**	**Mean**	**Sd**	**Min**	**Max**
Learning accuracy (%)	30	71.80	14.10	31.70	81.10
Self-confidence (%)	30	69.30	26.20	25	100
FMPS: concerns over mistake	26	21.20	7.66	12	43
FMPS: doubt about actions	26	11.00	3.34	5	18
FMPS: personal standards	26	22.00	4.82	11	34
Sensitivity to punishment	26	42.20	8.82	26	60
Sensitivity to rewards	26	36.10	6.75	21	52
State anxiety	26	33.60	9.85	20	55
Trait anxiety	26	40.90	12.30	21	67
Depression	26	5.00	3.62	0	14
Anxiety	26	6.38	3.72	2	17

Sd: standard deviation; min: minimum; max: maximum; FMPS: Frost Multidimensional Perfectionism Scale.

## Data Availability

All data are available upon request.

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
