# Peer review of "Impact of Metacognitive and Psychological Factors in Learning-Induced Plasticity of Resting State Networks"

_biology, 2022, doi:10.3390/biology11060896_

Round 1

Reviewer 1 Report

Impact of Metacognitive and Psychological Factors in Learning Induced Plasticity of Resting-State Networks

The authors aimed to study the effect of confidence in learning and psychological temperament on the adaptation of the resting state to experience s and stimuli.  Experimented on 35 adults, the authors observed that confidence in learning was positively associated with the degree of functional connectivity changes in 11 connections and the variability in functional connectivity was associated with an individual’s level of anxiety sensitivity. The study claims that functional connectivity changes related to metacognitive abilities differ from the changes linked to learning accuracy.

Review of abstract:

  1. Line 23 is misleading. The authors studied only ‘confidence in learning’ related functional connectivity changes. Based on this single parameter observation, claiming that FC reconfiguration at the resting state Is linked to metacognitive abilities is vague and misleading.
  2. There is a discrepancy between the count of subjects analyzed vs count of subjects experimented. Here the conclusion was drawn on the count of patients analyzed, but in abstract the number mentioned refers to the count of subjects experimented. It is strongly recommended to use the count of subjects analyzed.

General comments:

  1. Line 213: after excluding 2 participants, the sample size is still 35? or the total number of participants was 37 and after the exclusion, it was 35? Please use consistent and accurate subject counts.
  2. Line 244: Is the final subject count 29. If yes, can the authors make this number uniform across the manuscript? It is not necessary to include the subjects in the manuscript whose data was not used to perform the analysis or draw any conclusions.
  3. Line 249: Positive associations between ΔFC and learning accuracy were identified in six pairs of areas.
    1. What was the change in FC? Was it increased connectivity or decreased?
    2. How does the change look in control subjects?
    3. The positive association was measured using what algorithms/ methods?
    4. The authors mentioned 6 pairs of areas. What are the exact 6 pairs of areas they are talking about? Figure 1 in the supplement has more than 6 pairs of areas.
    5. All these connections concerned reinforcement between… What is the meaning of reinforcement?
    6. The cerebellum takes care of procedural memory, but the hippocampus takes care of the new memories. During the learning phase, did the authors observe any FC activation/ association with the hippocampus?
    7. The mentioned areas for learning accuracy network, do they get activated when other tasks are performed (probably authors should verify this from existing literature).
  4. How did the authors decide which one to mark the learning accuracy network?
  5. Line 256: How did the authors decide which FC to mark as a confidence network.
  6. Line 325: areas that were also identified in the present study. In the manuscript, there is no evidence of any FC measures with the hippocampus.
  7. The study lacks negative control.

Author Response

Thank you for your comments and for giving us the opportunity to resubmit the manuscript after revision. Here is the 2nd modified version in which we responded to all the reviewers’ comments and we have paid attention to better detail the methodology.

We responded to each one individually in this document and we highlighted the important updates in yellow in the manuscript.

See below our responses.

Reviewer number 1 :

Abstract :

  1. Line 23 is misleading. The authors studied only ‘confidence in learning’ related functional connectivity changes. Based on this single parameter observation, claiming that FC reconfiguration at the resting state Is linked to metacognitive abilities is vague and misleading.

We thank the reviewer for this comment; we agree that the present findings do not permit to specifically relate to metacognitive abilities. In consequences, we changed the term “metacognitive abilities” to “confidence in learning”.

  1. There is a discrepancy between the count of subjects analyzed vs count of subjects experimented. Here the conclusion was drawn on the count of patients analyzed, but in abstract the number mentioned refers to the count of subjects experimented. It is strongly recommended to use the count of subjects analyzed.

Effectively, we apologize for this miscommunication. 37 subjects were recruited, 35 subjects underwent the fMRI quality requirement, 30 subjects had completed the confidence rating and 26 subjects had completed all the psychological tests. In consequence, the rs analyses were based on 30 subjects and the association with psychological variables was based on 26 subjects.

We corrected it by specifying the number of subjects used in the rs anaylsis (30) in the abstract and in the method part.

General comments:

  1. Line 213: after excluding 2 participants, the sample size is still 35? or the total number of participants was 37 and after the exclusion, it was 35? Please use consistent and accurate subject counts.

Indeed, the total sample was 37 subjects. We clarified this point.

  1. Line 244: Is the final subject count 29. If yes, can the authors make this number uniform across the manuscript? It is not necessary to include the subjects in the manuscript whose data was not used to perform the analysis or draw any conclusions.

The final subject count is 30 subjects for the rs analyses and 26 for the associations with psychometrics variables. We clarified this point in the manuscript.

  1. Line 249: Positive associations between ΔFC and learning accuracy were identified in six pairs of areas.
    1. What was the change in FC? Was it increased connectivity or decreased?

Increased positive connectivity was found between the cerebellum 9 L and 9 R. All the others connectivity changes concern a decrease of positive connectivity. We specified it in the manuscript.

  1. How does the change look in control subjects?

There is no control group in this study, all the subjects who underwent the learning task were healthy.

  1. The positive association was measured using what algorithms/ methods?

Relationships between FC change and learning accuracy or confidence rating were examined using regression analyses directly implemented in the Conn Toolbox v18.b. We specified it in the method part.

  1. The authors mentioned 6 pairs of areas. What are the exact 6 pairs of areas they are talking about? Figure 1 in the supplement has more than 6 pairs of areas.

This concerns connection between :

  • Frontal inferior orbital L and Cerebellum vermis 4 5
  • Cerebellum 9 R and Amygdala R
  • Cerebellum 4 5 R and Lingual R
  • Cerebellum 3 L and Precentral R
  • Cerebellum 3 L and Parietal superior L
  • Cerebellum 3 L and Cerebellum 8 R

We sincerely apologize for the lack of clarity. We added the detailed results in the “results” section and we can propose another illustration if the reviewer considers it would be more appropriate.  

  1. All these connections concerned reinforcement between... What is the meaning of reinforcement?

We thank the reviewer for this comment which helped to improve our manuscript. We have changed, throughout the text, the term “reinforcement” with more precise and accurate adjectives defining the connectivity, ie. Increase, decrease or change.

  1. The cerebellum takes care of procedural memory, but the hippocampus takes care of the new memories. During the learning phase, did the authors observe any FC activation/ association with the hippocampus?

We have not yet explored the activation during the learning phase, but it will be investigated in a future analysis. However, we did find significant results concerning the hippocampus that displays connectivity change with the cerebellum linked to confidence rating. As discussed in the manuscript, the connectivity changes between the hippocampus and the cerebellum could arise from a history of co-activation during the learning task.

  1. The mentioned areas for learning accuracy network, do they get activated when other tasks are performed (probably authors should verify this from existing literature).

We thank the reviewer for this suggestion and we have completed the discussion part in consequences. Indeed, all the areas found in the learning accuracy network had already been involved in different kind of learning task; from classic conditioning to associative learning (Balsters et al., 2013; Edde et al., 2020; Molchan et al., 1994; Sehlmeyer et al., 2009).

We added this statement to the discussion part.

8.     How did the authors decide which one to mark the learning accuracy network?

The “learning accuracy” were identified in the same way as the “confidence network”, ie using regression analyses with connectivity changes in a whole brain exploratory analysis. Thus, areas belonging to the learning accuracy networks are connections in which changes of connectivity strength are linked to the % of accuracy during the learning task.

9.     Line 256: How did the authors decide which FC to mark as a confidence network.

Areas belonging to the confidence networks are connections in which changes of connectivity strength are linked to the % of confidence.

For both learning accuracy and confidence we specified this notion in the methods part.

10.  Line 325: areas that were also identified in the present study. In the manuscript, there is no evidence of any FC measures with the hippocampus.

The hippocampus has been identified in the confidence networks and was connected to the cerebellum. It is written in the results and displayed in the figure but we could highlight more these results if needed.

11.  The study lacks negative control. 

Again, we thank the reviewer for this remark. We had considered including a negative control; however, the task used may have tapped on a multitude of cerebral areas making it difficult to pick a particular region for controlling positive findings. This is also for this reason that we have decided to carry-out a whole brain exploratory analysis.

Reviewer 2 Report

This manuscript provides data related to impact of metacognitive and psychological factors in learning-induced plasticity. Below are my comments which authors may wish to address.

The age of participants recruited for this study was between 18-70 years. This age range is very wide. Learning can be impacted by age. Did the authors also analyzed whether ageing also affected their results. 

Authors should also include a table for other information about the participants. For example, years of education. This might also affect the learning and performance task. Because the training and experimental stimuli were different, any effect of education should be evaluated.

It is not clear how many older subjects were used in this study. The learning induced plasticity could be dramatically different in older adults.

 In the result section (page 6, line 244) a total of 29 participants (14 men; mean age 33.10 ± 13.4) were included due to missing data for certain self-report measures.

The mean age mentioned here is it for total 29 participants or only for 14 men?  Also, what are the missing data based on which only 29 participants were selected for correlation analysis? Were the self-reported measures obtained after experiments?

Also, the confidence was rated by the participants before the scan. Also, the anxiety level related to task and imaging may differ in different participants and could affect some of the parameters and also the confidence. Which makes the analysis and interpretation complex. This is also mentioned by the authors in the discussion.

Author Response

This manuscript provides data related to impact of metacognitive and psychological factors in learning-induced plasticity. Below are my comments which authors may wish to address.

  • The age of participants recruited for this study was between 18-70 years. This age range is very wide. Learning can be impacted by age. Did the authors also analyzed whether ageing also affected their results.
  • It is not clear how many older subjects were used in this study. The learning induced plasticity could be dramatically different in older adults.

Indeed, we mentioned that participants up to 70 years old could be recruited. In the present study, the large majority of the participants were younger adults (median = 29 years old) we have however one participant who is 60 years old and one participant who is 68 years old.

Concerning the effect of age on learning abilities, we checked its potential effect and did find significant negative correlation between age and % of learning accuracy.

It is also true that those participants could exhibit differences in learning-induced plasticity. As consequence, we conducted spearman correlation between the delta of connectivity within the networks (i.e., strength of changes) and age. Significant negative correlation was found between age and the delta of 5 out of the 11 networks, indicating that older participants tend to exhibit lower plasticity.

However, we did assess if our main result stands while controlling for the age effect. Thus, we conducted partial correlation analysis between delta of connectivity and anxiety sensitivity while controlling for age and education (see further details below).

3)Authors should also include a table for other information about the participants. For example, years of education. This might also affect the learning and performance task. Because the training and experimental stimuli were different, any effect of education should be evaluated.

We added education information in the population description.

After verification, we did find a significant association using Pearson correlations between the level of education and learning accuracy; and, higher education was related to higher learning accuracy. In a similar manner, higher education was linked to stronger changes in 2 out of the 11 confidence networks.

In consequence, we replicated our analysis while controlling for age and education simultaneously. The delta of connectivity of 7 out of 11 networks remained significantly correlated with anxiety sensitivity. We added accordingly these results in the manuscript.

In sum, we modified the manuscript to include the results concerning effects of age and education level; and, we further changed our analysis to include them as control variables.

  • In the result section (page 6, line 244) a total of 29 participants (14 men; mean age 33.10 ± 13.4) were included due to missing data for certain self-report measures. The mean age mentioned here is it for total 29 participants or only for 14 men? Also, what are the missing data based on which only 29 participants were selected for correlation analysis? Were the self-reported measures obtained after experiments?

The mean age concerns the total participants (men and women). As you will see in Table 1 (Description of participants), the missing data arise from the participants who did not complete the questionnaire on anxiety and depression, leading to 26 participants.

We clarified this point in the article.

The self-reported measures were obtained before the experiments which could lead to a bias during the learning task. As we cannot control for the examination effect, we included this limit in the “discussion” section.

Also, the confidence was rated by the participants before the scan. Also, the anxiety level related to task and imaging may differ in different participants and could affect some of the parameters and also the confidence. Which makes the analysis and interpretation complex. This is also mentioned by the authors in the discussion.

Thank you for these comments.
